# EEGs Disclose Significant Brain Activity Correlated with Synaptic Fickleness

**DOI:** 10.3390/biology10070647

**Published:** 2021-07-11

**Authors:** Jorge Pretel, Joaquín J. Torres, Joaquín Marro

**Affiliations:** Institute “Carlos I” for Theoretical and Computational Physics, University of Granada, E-18071 Granada, Spain; jorgepv@correo.ugr.es (J.P.); jmarro@ugr.es (J.M.)

**Keywords:** EEG time series, synaptic plasticity, modulations and explosive transitions in brain waves

## Abstract

**Simple Summary:**

In this study, we explore the emergence of oscillatory behavior similar to the signals of brain activity observed in electroencephalograms (EEGs) using a network of synaptic relations mingling excitatory and inhibitory neuron nodes. We identify abrupt variations on that activity brought about by swift synaptic mediations. These changes are originated by the slowdown of the activity of inhibitory neuron populations due to synaptic depression. The latter generates an imbalance between excitation and inhibition causing a quick explosive increase of excitatory activity, which turns out to be a (first-order) phase transition among different oscillatory states. Interestingly enough, near this transition, our model system exhibits oscillatory activity with a strong component in the *delta-theta domain* that coexist with fast oscillations and happens to be similar to the observed *delta-gamma* and *theta-gamma modulation* in actual brains. Our findings here help to understand actual brain activity data in terms of nonequilibrium phase transitions theory.

**Abstract:**

We here study a network of synaptic relations mingling excitatory and inhibitory neuron nodes that displays oscillations quite similar to electroencephalogram (EEG) brain waves, and identify abrupt variations brought about by swift synaptic mediations. We thus conclude that corresponding changes in EEG series surely come from the slowdown of the activity in neuron populations due to synaptic restrictions. The latter happens to generate an imbalance between excitation and inhibition causing a quick explosive increase of excitatory activity, which turns out to be a (first-order) transition among dynamic mental phases. Moreover, near this phase transition, our model system exhibits waves with a strong component in the so-called *delta-theta domain* that coexist with fast oscillations. These findings provide a simple explanation for the observed *delta-gamma* and *theta-gamma modulation* in actual brains, and open a serious and versatile path to understand deeply large amounts of apparently erratic, easily accessible brain data.

## 1. Introduction

Today one successfully associates most brain activity with events in which large sets of neurons cooperate mediated by an extensive range of variations of their synaptic relations [1]. These complex dynamic processes broadcast signals throughout, and EEG exploration on the cerebral cortex has thus become a relatively simple, convenient and inexpensive way of analyzing consequences of such an intriguing collaboration [2,3,4,5]. In fact, EEG studies deliver some overall image of the brain activity with good time accurateness that complements other exploratory methods of better spatial resolution, such as magnetic resonance imaging. Specifically, EEGs watch over frequencies and often distinguish δ,θ,α,β and γ “rhythms” —subsequently along the range 0.5 Hz to 35 Hz and more—which are loosely associated with different states of consciousness such as say, deep sleep, anesthesia, coma, relax, and attention, and to different mental and congnitive processes that the brain can perform [6].

Indeed, EEGs now provide a main noninvasive tool to deepen on the brain performance under both normal and pathological conditions [3,7,8,9,10]. Moreover, with the recent development of Machine Learning and Deep Learning techniques, EEG data have regained significance in recent years, as it has demonstrated to be of great convenience in the implementation of high-accuracy automated systems for detecting and diagnosing a broad range of neuropathologies [11], a task that generally requires the supervision of highly trained experts. It is therefore convenient to deepen further on the interpretation of all of those brain waves phenomena. Actually, several prototypes have already addressed the origin and nature of observed brain oscillatory behavior, e.g., [2,12,13,14,15]. Recently, following a hint [12] that α rhythms might come out from the filtering of cooperative signs by interactions with noisy signals from different parts of the nervous system, it was explained the emergence of a wide spectrum of brain waves within a simple computational framework [16]. More specifically, this study has shown that a neural module can exhibit waves in a variety of frequency bands just by tuning the intensity of a noisy input signal. We interpret this result as suggesting the existence of a possible unifying mechanism for a range of oscillations. In fact, existing literature by now has described [1] various well-defined (let us say) *dynamic phases*, as well as transitions among them—typically, from states with a low and incoherent activity to others that show a high synchrony—where weak signals are processed efficiently despite the presence of significative levels of noise. This potential faculty of neural networks has its origins in a large susceptibility developed in the vicinity of a phase transition, due to a mechanism generally known as *stochastic resonance* [17].

The picture in previous theoretically oriented and computational EEG works, including [16], is mostly phenomenological and generally adopts a uniform and stationary description of the neuron relations efficiency, thus forgetting the actual possibility that synapses perform dynamically during the neuron cooperation processes [1,18,19,20]. Nevertheless, synaptic relations certainly vary with time at short time scales while affecting essentially both the neuron network global behavior and the ensuing capacities to transmit information [1,20,21,22,23,24,25]. It was reported, for instance, that such short-term synaptic plasticity may induce in the human cortex bursting followed by asynchronous activity [20], as well as instabilities prompting transitions among the attractors of the network dynamics, which allow for effective memory searching [25], in addition to a form of ‘up-and-down states’ reported to occur in cortical neuron populations [26]. Additionally, a sort of sudden synaptic facilitation can allow for transient persistent activity after the removal of a stimulus [24], which may be the basis for working memory.

Aware of these and similar facts, we mathematically recast and generalize here both the more mesoscopic description in [2] and the algorithmic model in [16], perfecting them with detailed dynamic synapses and other realistic features including more random topologies (see additional study in Appendix A). We thus show how certain levels of short-term depression of the synaptic links induce transitions between states of synchronized activity in excitatory and inhibitory neuron populations and global states of incoherent behavior. It follows that one may speak of kind of sharp phase transitions, clearly displaying metastability and hysteresis, that have been experimentally observed [27]. Furthermore, near such “explosive transitions”, our model exhibits oscillations with a prominent component in the δ−θ band along with high frequency activity, namely the δ−γ and θ−γ modulations already perceived in actual brain EEG recordings [28,29], which have been associated with *fluid* intelligence [30]. Even more, we here associate such intriguing sharp variations with disruptions of the balance among excitation and inhibition produced by depression of excitatory inputs arriving to inhibitory neurons. This reduces the inhibitory activity thus prompting a sudden excitation increase that further reduces inhibition. Interestingly, enough, a lack of the excitation-inhibition balance in the actual human brain could be crucial to understand the essentials of some recurrent neurological disorders such as epilepsy, autism and schizophrenia, e.g., [31,32].

Moreover, the present EEG model constitutes a realistic, highly configurable neural network capable of reproducing a wide range of dynamical behaviors akin to those observed in actual measurements in normal and pathological conditions, and could also be used to detect phase transitions in actual EEG recordings. We believe that further work along the present lines will surely benefit a much more useful interpretation of (easily available) EEG data.

## 2. Materials and Methods

The simplest version of our model aims to capture the essentials of the cerebral cortex operation allowing for a network with excitatory (E) and inhibitory (I) neurons, the former occurring four times the latter. Furthermore, the amplitude of the corresponding postsynaptic responses follow the opposite ratio, i.e., the response evoked by any I is four times larger than that by any E. This is supposed to correspond to a realistic *cortex balanced state* [33,34]. We then represent a region of the cerebral cortical tissue with a large 2-dimensional square of *N* nodes with periodic boundary conditions and fulfilling such a balance, in which each I node influences a set of 12 neighboring E’s and it is influenced by 32 adjacent E’s as illustrated in Figure 1 Left. As in previous work [12,16], we will not consider here E-E or I-I feedbacks, in fact, preliminary simulations of our system show that including local recurrent excitatory or inhibitory connectivity does not significantly alter our main results (see the additional study in Appendix A). Moreover, from the various familiar types of existing neuron dynamics, we refer to the celebrated integrate-and-fire case [1,35]. Specifically, the cell membrane acts as a capacitor subject to several currents, which results in a potential *V* for each neuron changing with time according to.
(1)$$\tau \frac{\mathrm{dV}}{\mathrm{dt}}=-V+V^{in}+V^{\mathrm{noise}}.$$

Here, as in previous works [12,16], the time constant τ is set equal to $\tau_{1}$ ($\tau_{2}$) when the membrane cell voltage is above (below) a certain resting potential, which we set to zero. The last two terms in (1) correspond to the voltage induced by the sum of all currents through the membrane, which we separate here in two main contributions. $V^{in}$ is the sum of inputs from adjacent neighbors that influence the given cell, while $V^{\mathrm{noise}}$ accounts for any input from neurons in other regions of the brain. Assuming lack of correlations [36], we represent $V^{\mathrm{noise}}$ as a Poisson signal characterized by a noise level parameter *μ.*

It is well established that in human brains, synapses linking neurons may undergo variations in scales from milliseconds to minutes, in addition to more familiar long-term plastic effects. In particular, one observes short-term depression (STD), in which the synaptic efficacy decreases due to depletion of neurotransmitters inside the *synaptic button* after heavy presynaptic activity [18]. In addition, the presence of a short-term facilitation process is generally reported, characterized by an increase of the efficacy strength [37,38,39], which results from a growth of the intracellular calcium concentration after the opening of the voltage gated calcium channels due to successive arrival of action potentials to the synaptic button. It seems that in general, these two short-term mechanisms may compete [1,23] but, for simplicity, we only consider here synapses endowed of STD, and describe this using the release probability *U* and the fraction of neurotransmitters at time *t* ready (to be released) after the arrival of an action potential $x_{t}$ [22]. In this case, each time a presynaptic spike occurs, a constant portion *U* of the resources $x_{t}$ is released into the synaptic gap, and the remaining fraction $1-x_{t}$ becomes available again at rate $1/\tau_{\mathrm{rec}}$. Therefore,
(2)$$\frac{{\mathrm{dx}}_{t}}{\mathrm{dt}}=\frac{1-x_{t}}{\tau_{\mathrm{rec}}}-Ux_{t}\delta \left(t-t_{\mathrm{sp}}\right)$$
where the delta function imposes that the second right-hand term only occurs for $t=t_{\mathrm{sp}}$, the time at which a presynaptic input spike arrives. Assuming also the amplitude of the response to be proportional to the fraction of neurotransmitters released after the input spike, the STD effect can be expressed, for E and I neurons respectively, as follows
(3)$$V_{t}^{in,d}=V_{0}^{d}Ux_{t_{\mathrm{sp}}}\left[\Theta \left(t-t_{\mathrm{sp}}\right)-\Theta \left(t-t_{\mathrm{sp}}-t_{\max}\right)\right]$$
(4)$$V_{t}^{in,h}=V_{0}^{h}Ux_{t_{\mathrm{sp}}}\Theta \left(t-t_{\mathrm{sp}}\right)e^{\frac{-(t-t_{\mathrm{sp}})}{\tau_{2}}}$$
where Θ(X) denotes the Heaviside step function. The form of these inputs generated by E and I neurons are chosen so that the response produced on the postsynaptic neuron membrane matches data; see, for instance, [16]. Thus, for simplicity, we model the excitatory synaptic input by a square pulse of width $t_{\max}$ and maximal amplitude $V_{0}^{d}$, as described by Equation (Equation 3), and the inhibitory input with a maximum amplitude $V_{0}^{h}$ followed by an exponential decay with time constant $\tau_{2}$, as in Equation (Equation 3). In addition, to account for synaptic strength variations that depend on presynaptic history, we multiply these input functions by a factor $U\cdot x_{t_{\mathrm{sp}}}$, thus ensuring that the amplitude of the synaptic input is proportional to the amount of neurotransmitters released right after a presynaptic spike, which is an activity dependent factor through dynamics in Equation (Equation 2). Please note that there is no synaptic variability present when $U\cdot x_{t_{\mathrm{sp}}}=\mathrm{constant}$ occurring for $\tau_{\mathrm{rec}}\to 0.$ Furthermore, to prevent the membrane potential in (1) from reaching physiologically unrealistic levels, we impose upper and lower limits of $V_{\mathrm{sat}}=90$ mV and $V_{\min}=-20$ mV, respectively, around the resting membrane potential, $V_{\mathrm{rest}}=0$ as said. This is achieved by multiplying the different excitatory and inhibitory inputs by the terms $\left(V_{\mathrm{sat}}-V\right)/V_{\mathrm{sat}}$ and $\left(V_{\min}-V\right)/V_{\min}$, respectively.

Equations (Equation 1)–(Equation 4) fully describe the dynamics of the membrane potential in our basic model below a firing threshold, which is in principle set $V_{\mathrm{th}}=6\mathrm{mV}$ above the resting membrane potential for both E and I neurons. Additionally, after the generation of a spike at $t_{f}$, we assume an absolute refractory period ($t_{a}$) during which the neuron is unable to fire again, and a subsequent relative refractory in which the ability to produce new spikes is constrained. Therefore, we set
$$V_{\mathrm{th}}\left(t\right)=\left\{\begin{array}{c} V_{\mathrm{sat}}t_{f}<t<t_{f}+t_{a}\\ 6+\left(V_{\mathrm{sat}}-6\right)e^{-\kappa \left(t-t_{f}-t_{a}\right)}t>t_{f}+t_{a}. \end{array}\right.$$

That is, the threshold is first set to $V_{\mathrm{sat}}$ (during 100 time steps, which gives $t_{a}=4\mathrm{ms}$) to impede any further spike generation during $t_{a}$. Then, it decays exponentially to its resting value of 6 mV with a time constant $\kappa^{-1}=0.5\mathrm{ms}$ that mimics the existence of a relative refractory period.

## 3. Results

In absence of STD, the present model has previously been successful to accurately reproduce relevant features of different type of brain waves in actual EEG recordings as, for instance, the power spectrum of α-waves in thalamus and their steady-state voltage distribution (see Refs. [12,16]). Moreover, we already reported in [16] a unifying framework for the generation of different brain rhythms just by tuning the level of uncorrelated excitatory inputs that a cortical neural population received from other brain areas, reproducing oscillatory behavior in the α, β, γ and ultrafast bands, as observed in actual EEGs.

Using this clear-cut realistic model, we numerically analyzed how synaptic STD affects eventually emergent waves by carefully monitoring the network dynamics for adiabatically increasing values of the noise parameter *µ*. Figure 2 depicts the resulting average membrane potential of the E population versus *µ*, which clearly illustrates the mentioned sharp transitions, i.e., in the absence of STD (top panel in each column), the well-defined nature of waves does not vary with the external noise amplitude within *µ*∈ (0.5, 100), as already reported in [16]. This regime corresponds to the simplest and most familiar brain waves. However, when STD is on—specifically, when the parameter $\tau_{rec}$ is large enough so that synaptic efficacies vary noticeably with system activity—an ‘explosive’ transition may show up as *µ* increases. This occurs at lower $\tau_{\mathrm{rec}}$ the lower the maximal excitatory postsynaptic amplitude $V_{0}^{d}$ is. It is said ‘explosive’ in the sense that the transition shows hysteresis, from well-defined synchronized behavior to a state of high excitation and low coherence, as we vary *µ* adiabatically forwards (purple line) and backwards (green line). One may also name this a *first-order phase transition* by simple analogy with thermodynamics, though with the warning that the present setting is a nonequilibrium one [40]. These findings, i.e., the explosive transition and the emergence of hysteresis, are robust to size increases (see additional analysis in Appendix A), so it is, therefore, likely to occur even for populations with a considerable number of neurons.

The resulting phase diagram in the $\left(\mu ,\tau_{\mathrm{rec}}\right)$ space is illustrated in Figure 3. The solid quasi-vertical line, for μ≈0.5, describes a (continuous or second-order) phase transition between a near silent phase A, with asynchronous sporadic spikes at low rate (corresponding to the asynchronous down state actually observed in the brain), and an oscillatory phase B, where brain waves emerge with increasing frequency as μ increases (see Figure 2). As $\tau_{\mathrm{rec}}$ increases in the system, Figure 3 indicates that the brain waves disappear at a (first-order) transition (dashed line), where a new phase D of waves with high excitation and low coherence emerges. This sharp transition becomes smooth above a say ‘tricritical point’ (1.4, 268) (short quasi-vertical solid line on top). The small region C shows metastability as revealed by hysteresis. Please note that when μ is large this region C narrows as noise level increases. In addition, region B contains (red and blue) areas in which brain waves sharply emerge with high values of the firing rates (>100 Hz) for E and I neurons.

Trying to deepen the understanding of the sharp transition, here we monitored (Figure 4) the change with the level of synaptic depression of both the mean firing rate and the mean amplitude of the oscillations in E and I neuron populations (for *µ*= 3). This shows that as STD increases, a decline in the excitatory efficacy induces I neurons to decrease their activity as the system approaches the transition point, where they become silent. A feedback induced by this decay of the I activity promotes the E’s firing activity until reaching (at the transition point) its maximum possible, from where they remain firing tonically. This induces important effects on the ensuing oscillations: the average membrane potential of the inhibitory neurons jumps to zero at the transition point (corresponding to a complete absence of firing), and the average membrane potential of the Es neurons decays to a very low value below $V_{\mathrm{th}}.$

Additionally interesting is how the nature of the emerging waves changes with STD. For a relatively low noise, e.g., *µ* = 0.8, the network’s response remains nearly unchanged, while the amplitude of the oscillation decays until no well-defined oscillatory behavior is observed as STD is increased (Figure 5, case *µ* = 1). Please note that this transition from a state with synchrony to an incoherent one become abrupt as described above for a level of noise μ>1 (see Figure 5).

For higher values of *µ* (e.g., μ=3 in Figure 6), the power spectrum of the response shows significant changes. First, its peak frequency notably increases for higher levels of STD, becoming up to twice as great as for the static case ($\tau_{\mathrm{rec}}=0$), thus, inducing waves in the β and γ regimes. This STD-induced transition from low to high frequency bands confirms that synaptic plasticity could be an important mechanism involved in modulating the nature of the oscillations from cortical neuronal populations. In addition, we observe that an increase of $\tau_{\mathrm{rec}}$ can produce secondary, low-frequency peaks coexisting with the main peak in the power spectrum of the emergent waves. This phenomenon is most evident near the explosive transition point, where a prominent component in the δ/θ bands emerges, accompanied by a general enhancement in the amplitude of the oscillations, as can be seen in the time series presented in Figure 6. This effect seems to occur for all relatively high levels of noisy, namely μ>1.

Concerning the effect of the E/I balance on emergent behavior, it is of interest to study how it affects the incidence of $\delta \left(\theta \right)-\gamma$ modulations around the transition, and how the appearance of this is affected by the level of synaptic depression. Figure 7 illustrates some effects of changing the ratio between the E and I synaptic efficacies. We observe that when $V_{0}^{d}/V_{0}^{h}$ decreases and the inhibitory synapses become relatively more influential, the low-frequency δ/θ component becomes more significant for oscillatory behavior (see Figure 7, top-right and bottom-left panels) while when this ratio increases, the low frequency band components (δ and θ) tend to disappear (Figure 7, top-left and bottom-right panels). Additionally, an increase of $V_{0}^{d}/V_{0}^{h}$, which implies more excitation, makes the oscillations frequency more susceptible to changes on synaptic depression (see Figure 7, top-left and bottom-right panels), while a stronger inhibition tends to maintain the frequency of the emergent waves nearly unchanged against depression increases (Figure 7, top-right and bottom-left panels), thus leading to a homeostatic effect.

## 4. Conclusions

Summing up, we present in this work a very simple computational model of brain waves that, recasting previous EEG related work, has two significant features. One is that it provides a well-defined set-up to undertake a systematic study and interpretation of apparently erratic brain EEG data. These are easily accessible today and, as suspected (according to our outline here), happen to carry important information concerning the brain’s activity. Furthermore, this model is convenient to admit complements that one may expect to be relevant in these scenarios such as, for instance, other synaptic mechanisms, complex synaptic networks and more realistic neurons. In addition, and perhaps even more transcendental within this context, the framework presented here precisely illustrates how the concept of a (nonequilibrium) phase transition [40] may be essential for an accurate description of the brain properties. Lastly, the EEG model here presented constitutes a realistic, highly configurable neural network capable of reproducing a wide range of dynamical behaviors akin to those observed in actual measurements, and could therefore be of use in the task of training artificial neural networks for extracting features from brain activity in normal and pathological conditions.

## Figures and Tables

**Figure 1 biology-10-00647-f001:**
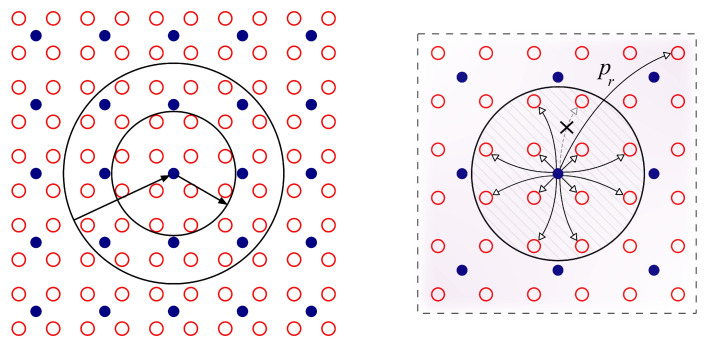
Schemes representing the network topologies used in the present study. (**Left**) The standard network topology used for the generation of brain waves in a E/I balanced cortical neuron population module as in Refs. [12,16]. The neuron coupling scheme represented in the panel is such that the small circle contains 12 E neurons (open red circles) that receive synaptic inputs from the inhibitory neuron (filled blue circle) in its center. On the other hand, the large circle contains 32 E neurons that excite the inhibitory neuron in its center. (**Right**) To see the robustness of the reported results in our study we also use a more complex small-world topology rewiring the given E-I and I-E connections in the network with some rewiring probability $p_{r}$ (see Appendix A for the associated study). In the illustrated example in the figure, one of the inhibitory connections of an I neuron to a given E neuron within its influence circle (as explained in the left panel) is exchanged with probability $p_{r}$ by a new inhibitory connection into other excitatory neuron in a random position in the network.

**Figure 2 biology-10-00647-f002:**
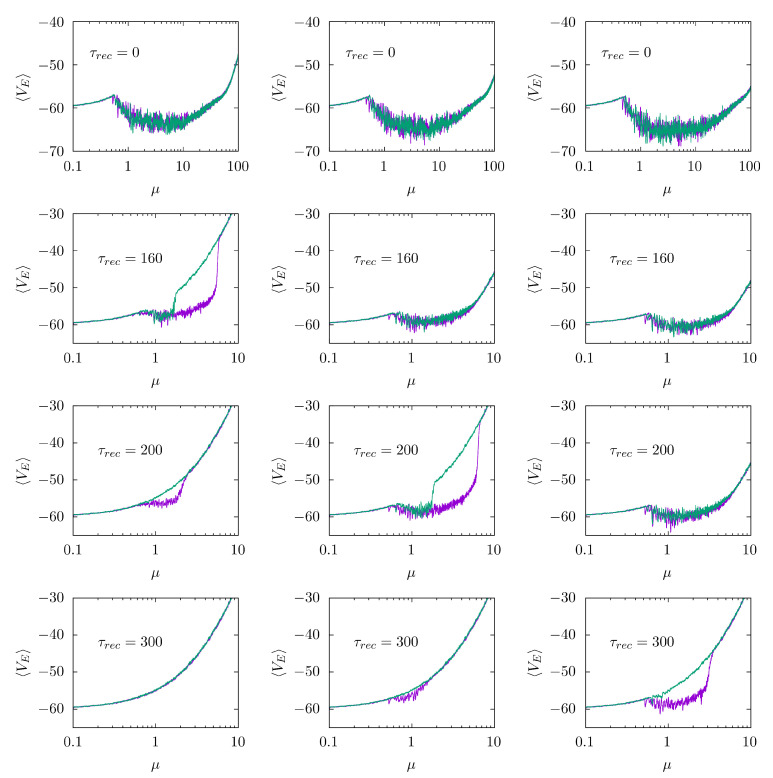
Evidence for sharp changes in emergent EEG-like waves as the noise level *µ* varies adiabatically and when synaptic depression is set on. Purple line corresponds to the adiabatically forward (positive) variation of the noise level *µ* whereas green line corresponds to a backward (negative) adiabatic variation. Columns correspond, from left to right, to $V_{0}^{d}=8,10$ and 12 mV, respectively. In all cases $V_{0}^{h}=-4V_{0}^{d}$, $U=0.5,\tau_{1}=16$ ms, $\tau_{2}=26$ ms and the amplitude of the external depolarizing inputs is $V_{o}^{d}=5.48$ mV. This corresponds to a module with 196 E’s and 49 I’s.

**Figure 3 biology-10-00647-f003:**
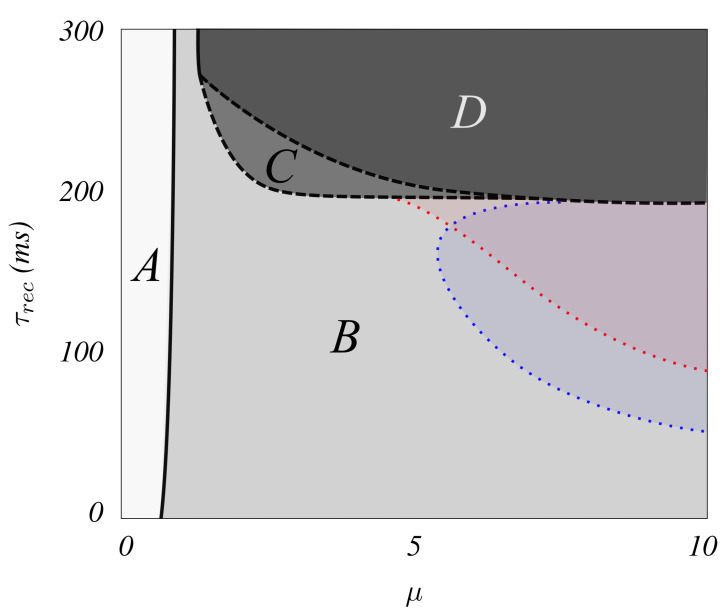
Diagram $\left(\mu ,\tau_{\mathrm{rec}}\right)$ illustrating the different dynamical phases in our system. For low noise (region A), random sporadic excitatory firing events emerge, which are unable to depolarize the I neurons. Region B shows well-defined rhythms ranging from α to γ bands, while a higher depression induces a cease of the inhibitory activity and a consequent absence of synchronicity and coherence in region D. Metastability, as illustrated in Figure 2, characterizes the region C. Red and blue colored areas in B indicate emerging waves with high values of the firing rates (>100 Hz) for E and I neurons, respectively. Dashed lines illustrate first-order phase transitions, while continuous lines denote second-order transitions.

**Figure 4 biology-10-00647-f004:**
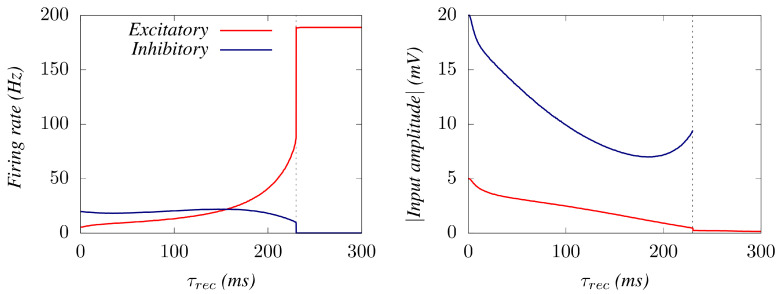
(**Left**) Average firing rates for E and I neurons as the level of STD is increased until the explosive transition occurs, for an external depolarizing noise level μ=3. (**Right**) Corresponding average synaptic current amplitudes $\langle |U\cdot x_{t_{\mathrm{sp}}}|\rangle$. This illustrates how the transition occurs because of a cease of firing of I’s due to the heavy synaptic depression of E’s. Beyond the transition point, excitatory neurons fire ceaselessly, further increasing their synaptic depression level.

**Figure 5 biology-10-00647-f005:**

Emergence of “α rhythms” (around 10 Hz) in the model for noise levels *µ* = 0.6, 0.8 and 1.0, respectively, from left to right. Although the power of the main frequency peak of the waves decays as STD increases, the case of *µ* = 1.0 clearly illustrates how these waves details are not dramatically affected by synaptic depression for this noise level and the α band regime remains until $\tau_{\mathrm{rec}}\approx 260\;\mathrm{ms}$, where the waves disappear (note that this transition becomes explosive for μ≳1 as shown in Figure 6).

**Figure 6 biology-10-00647-f006:**
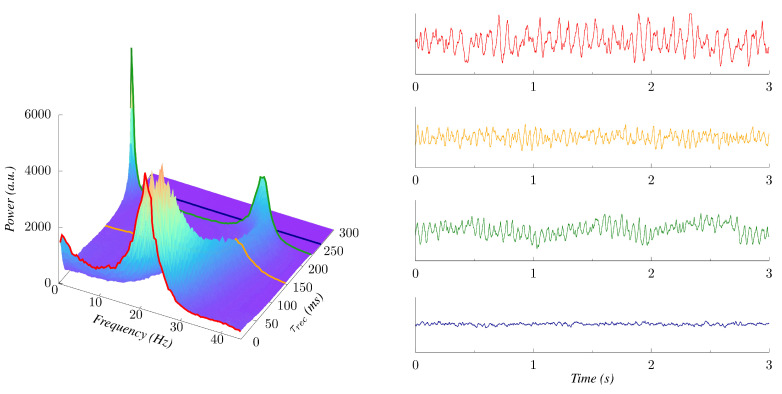
(**Left**) Power spectra of the system response as a function of the recovery time $\tau_{rec}$ for μ=3. (**Right**) time series of the emergent oscillations for particular levels of synaptic depression, namely $\tau_{rec}$ = 0, 145, 230 and 265 ms, respectively, from top to bottom. The associated power spectra for each of these series are highlighted (with the same color) in the surface plot of the left panel.

**Figure 7 biology-10-00647-f007:**
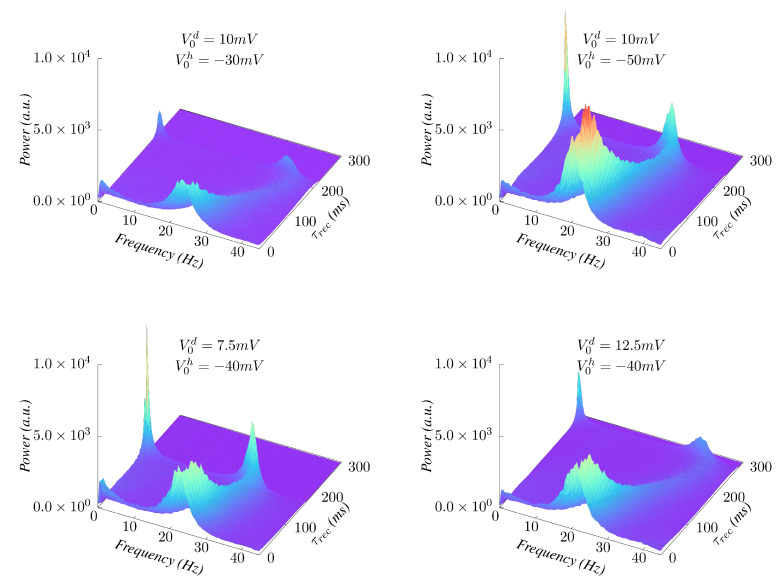
Effect of varying the E/I amplitude ratio as depression increases. (**Top**) Increasing the amplitude of I’s while leaving the E’s unchanged enlarges the δ/θ component of the δ(θ)−γ modulation around the phase transition ($\tau_{\mathrm{rec}}\approx 230$ ms). (**Bottom**) Increasing the E’s while maintaining the I’s moves the transition point to higher levels of depression and makes the frequency of the emergent oscillations more sensitive to synaptic depression.

## Data Availability

All data that support this study is available in the manuscript.

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
