# Peer review of "EEGs Disclose Significant Brain Activity Correlated with Synaptic Fickleness"

_biology, 2021, doi:10.3390/biology10070647_

Round 1

Reviewer 1 Report

In this work a simple model that displays oscillations similar to electroencephalogram brain waves is studied.

The model consists of a network of synaptic relations mingling excitatory and inhibitory neuron

nodes recast and generalized from an algorithmic model recently published and co-authored by two of the co-authors of this manuscript. The main change from the previous model is that short-term depression, in which the synaptic efficacy decreases due to depletion of neurotransmitters inside the synaptic button after heavy presynaptic activity is considered in a dynamic way. Abrupt variations caused by the slowdown of the activity in neuron populations due to synaptic restrictions are obtained. The consequent imbalance between

excitation and inhibition produces a quick explosive increase of excitatory activity, that is associated to a first-order transition among dynamic mental phases. Waves with a strong component in the delta-theta domain coexisting with fast oscillations are found near this phase transition. This result may explain the observed delta-gamma and theta-gamma modulation in actual brains, and of the large amounts of apparently erratic, brain data.

The paper is written in a clear way and deserve dissemination. Thus, I strongly recommend acceptance. 

Author Response

Reviewer I:

The reviewer says:

In this work a simple model that displays oscillations similar to electroencephalogram brain waves is studied.

The model consists of a network of synaptic relations mingling excitatory and inhibitory neuron nodes recast and generalized from an algorithmic model recently published and co-authored by two of the co-authors of this manuscript. The main change from the previous model is that short-term depression, in which the synaptic efficacy decreases due to depletion of neurotransmitters inside the synaptic button after heavy presynaptic activity is considered in a dynamic way. Abrupt variations caused by the slowdown of the activity in neuron populations due to synaptic restrictions are obtained. The consequent imbalance between excitation and inhibition produces a quick explosive increase of excitatory activity, that is associated to a first-order transition among dynamic mental phases. Waves with a strong component in the delta-theta domain coexisting with fast oscillations are found near this phase transition. This result may explain the observed delta-gamma and theta-gamma modulation in actual brains, and of the large amounts of apparently erratic, brain data. 

The paper is written in a clear way and deserve dissemination. Thus, I strongly recommend acceptance. 

Our response:

We very much appreciate the reviewer positive assessment of our manuscript. In our new, revised version we tried to improve quality while including the suggestions of the two other reviewers. We also improved the readability of the manuscript, checking extensively the English language and correcting some typos.

Reviewer 2 Report

This interesting article addresses an important issue – impact of low-frequency and high-frequency variations of synaptic plasticity on the behavior of neural networks. Results and methods reported in the article continue the line of inquiry that was initiated earlier by one of the authors (J. J. Torres)  and his colleagues, and is focused on the emergent characteristics of networks with ‘dynamic synapses.’ The modeling techniques are mathematically sound, theoretical suggestions associating sharp (explosive) changes in the patterns of excitatory-inhibitory activities in the model with some observable modulations in EEG patterns in the brain,  and the subsequent suggestions regarding manifestations of those patterns in mental mechanisms (e.g., working memory) appear to be justified and intuitively appealing.

Regrettably, the quality of the text  in this article does not correspond with the quality of ideas and potential significance of the reported findings. Low quality presentation interferers with comprehending the argument  and is likely to diminish the impact of the paper. This reviewer recommends re-organizing and re-writing the article:

  1. Perhaps, introduction can describe the mental phenomena of interest (e.g., working memory), followed by a few paragraphs mapping the phenomena onto the properties of networks with ‘dynamic synapses’ and relating those properties to EEG patterns. Such introduction will help grasping the overall idea and will motivate following details in the rest of the paper.
  2. It could be helpful to have a section examining similarities (and dissimilarities) between activity patterns in the model and observable EEG patterns.
  3. Definitely, extensive editing of English language and style are required. 

Author Response

Second reviewer

The reviewer says:

This interesting article addresses an important issue – impact of low-frequency and high-frequency variations of synaptic plasticity on the behavior of neural networks. Results and methods reported in the article continue the line of inquiry that was initiated earlier by one of the authors (J. J. Torres)  and his colleagues, and is focused on the emergent characteristics of networks with ‘dynamic synapses.’ The modeling techniques are mathematically sound, theoretical suggestions associating sharp (explosive) changes in the patterns of excitatory-inhibitory activities in the model with some observable modulations in EEG patterns in the brain,  and the subsequent suggestions regarding manifestations of those patterns in mental mechanisms (e.g., working memory) appear to be justified and intuitively appealing.

Our response

We very much appreciate the reviewer positive assessment of the models, analysis and results reported in our manuscript.

This reviewer also says:

Regrettably, the quality of the text does not correspond with the quality of ideas and potential significance of the reported findings. Low quality presentation interferers with comprehending the argument and is likely to diminish the impact of the paper. This reviewer recommends re-organizing and re-writing the article:

  1. Perhaps, introduction can describe the mental phenomena of interest (e.g., working memory), followed by a few paragraphs mapping the phenomena onto the properties of networks with ‘dynamic synapses’ and relating those properties to EEG patterns. Such introduction will help grasping the overall idea and will motivate following details in the rest of the paper.
  2. It could be helpful to have a section examining similarities (and dissimilarities) between activity patterns in the model and observable EEG patterns.
  3. Definitely, extensive editing of English language and style are required. 

Our response:

Following the reviewer suggestions, we have improved the quality of the text and English grammar throughout the manuscript. Also, we have reorganized the introduction and the results section.

On the other hand, concerning point 1 above we think the reviewer has misinterpreted some of our results: we are not focusing our study in explaining any specific mental process such, for instance, working memory. However, as it is now clear in the text, it is true that one could in principle associate specific mental processes to different types of waves at each phase, which would require a much more elaborated and complex model and serious further studies of it. We hope this will certainly be made in the near future (by us or by others) at the light of specific EEGs studies. On the other hand, following the reviewer suggestion we now include further comments on dynamic synapses and their relation for instance with working memory mental processes in the introduction section.

Concerning point 2, the standard model for generation of brain waves we are using has shown to accurately reproduce relevant features of different type of brain waves in actual EEG recordings as, for instance, the power spectrum of alpha waves in thalamus and their steady-state voltage distribution (see references [11] and [15]). Moreover, in [15], we already reported a unifying framework for the generation of different brain rhythms just increasing the level of uncorrelated excitatory inputs that a cortical neural population can receive from other areas, and we were able to generate alpha, beta, gamma and ultrafast oscillations as observed in actual EEGs. On the other hand, the inclusion of dynamic synapses in the model does not dramatically affect the features of the emerging rhythms (see for instance our new figure 2), but new phenomena can appear as an explosive phase transition when the level of synaptic depression increases. Then, we can clearly state that the present model framework can accurately reproduce oscillatory behavior similar to actual brain rhythms as those observed in EEG recordings. However, as said above, to reproduce the particular activity patterns associated with actual mental processes or tasks, requires to make a more detailed model study of such experimental task that we think is beyond the scope of the present study.

Following the suggestion of the reviewer, in the revised version of the manuscript we reorganized the result section adding a paragraph at the beginning reporting how the present model has been successful  to reproduce features of brain rhythms in actual EEGs.

Reviewer 3 Report

The authors present an iteration on their system model of EEG waves and incorporate the effect of short-term depression of synaptic efficacy. The topic is quite interesting and the authors do a good job of explaining their new findings. 

The paper could be improved in a few ways. This reviewer had to read their prior paper [15] to fully understand this paper.

1) The methods section needs to be expanded to discuss how the neurons are connected together (topography) like in the previous paper. 

2) Figure 1 needs to have the description of the different colored curves in the figure caption.

3) The font in the axes label etc. in figures 1 and 4 needs to be increased.

4) some of the English grammar could be improved. For example, on line 181, "Trying to deep on the nature".... does not make sense to me. Trying the deepen the understanding of the nature...  is more correct.  

Author Response

Third reviewer:

The reviewer says:

The authors present an iteration on their system model of EEG waves and incorporate the effect of short-term depression of synaptic efficacy. The topic is quite interesting and the authors do a good job of explaining their new findings.

Our response:

We very much appreciate the very positive reviewer assessment of our manuscript

 This reviewer also says

The paper could be improved in a few ways. This reviewer had to read their prior paper [15] to fully understand this paper.

  • The methods section needs to be expanded to discuss how the neurons are connected together (topography) like in the previous paper. 
  • Figure 1 needs to have the description of the different colored curves in the figure caption.
  • The font in the axes label etc. in figures 1 and 4 needs to be increased.
  • some of the English grammar could be improved. For example, on line 181, "Trying to deep on the nature".... does not make sense to me. Trying the deepen the understanding of the nature...  is more correct.  

Our response:

In the revised version of the manuscript we have incorporate the reviewer suggestions and now we explain in the method section how is the network topology of our system. We also improved the quality of the figures including the reviewer suggestions, that is, a new figure to explain the network topology (new figure 1) increasing the labels of figures 1 and 4 (new figures 2 and 5) and describing the coloured lines in figure 1 (new figure 2). Finally, we have rewritten some parts of the manuscript improving the English grammar and correcting the existing typos.

Round 2

Reviewer 2 Report

Thank you for the changes and corrections - the paper is more coherent now and reads better. Clarifications and details have been introduced that, I believe, are useful and were missing in the previous version. 

I would still recommend grammar checking and careful  editing.  For example, Figure 1 cuts into a sentence, separating text from the equation it introduces.